# Computational Modeling of Low-Abundance Proteins in Venom Gland Transcriptomes: *Bothrops asper* and *Bothrops jararaca*

**DOI:** 10.3390/toxins17060262

**Published:** 2025-05-22

**Authors:** Joseph Espín-Angulo, Doris Vela

**Affiliations:** 1Facultad de Hábitat, Infraestructura y Creatividad, Pontificia Universidad Católica del Ecuador, Quito 170525, Ecuador; josephahrim@gmail.com; 2Laboratorio de Genética Evolutiva, Facultad de Ciencias Exactas, Naturales y Ambientales, Pontificia Universidad Católica del Ecuador, Quito 170525, Ecuador

**Keywords:** Alphafold2, conserved domains, bioinformatics tools, CHIMERA, toxins

## Abstract

Snake venoms contain numerous toxic proteins, but low-abundance proteins often remain uncharacterized due to identification challenges. This study employs a bioinformatics approach to identify and structurally model low-abundance proteins from the venom gland transcriptomes of *Bothrops asper* and *Bothrops jararaca*. Using tools such as tblastn, Jalview, and CHIMERA, we analyzed sequences and structural features of proteins including arylsulfatase, CRISP (Cysteine-Rich Secretory Protein), von Willebrand factor type D (vWFD), and dihydroorotate dehydrogenase (DHODH), and identified potential new isoforms of SVMP-PIIIb (Ba_1) and botrocetin in *B. asper*. Protein models were generated with AlphaFold2, compared with crystallized structures from the Protein Data Bank (PDB), and validated using Procheck, ERRAT, and Verify3D. Conserved motifs and domains were annotated through Pfam and InterPro, revealing structural elements that suggest possible roles in venom physiology and toxicity. These findings emphasize the potential of computational biology to characterize structurally relevant but experimentally inaccessible venom proteins, and to lay the groundwork for future functional validation.

## 1. Introduction

Snake venom is a complex mixture composed primarily of enzymatic and non-enzymatic proteins, as well as small peptides [1,2]. The pathological effects produced in prey depend on the type of venom, which generally causes neuromuscular paralysis, hemorrhages, coagulopathies, swelling, and necrosis [3,4]. The most recognized families of venomous snakes include the Elapidae, Viperidae, and the subfamily Atractaspidinae [1,3,5]. Viperid snakes, such as those belonging to the genus *Bothrops*, possess hemotoxic venom, whereas elapids, such as those in the genus *Naja*, produce peptides with neurotoxic effects [1,6].

Advancements in sequencing technologies have enabled deeper exploration into the molecular composition of snake venoms. The most extensively studied venom proteomes and transcriptomes include those of *Bothrops jararaca* [7], *Echis coloratus* [8], *Ophiophagus hannah* [9], *Bothrops moojeni* [10], and *Azemiops feae* [11]. Several studies have shown that viperid venoms are largely composed of phospholipases, metalloproteinases, serine proteases, and C-type lectins, whereas elapid venoms are dominated by three-finger toxins (3FTxs) and phospholipases A2 [9,11,12,13].

In Central and South America, snakes of the genus *Bothrops* are responsible for the majority of snakebite envenomations, with *Bothrops asper* and *Bothrops jararaca* being notable for their hemolytic venom [13,14]. The venom of these species largely consists of metalloproteinases, phospholipases A2, serine proteases, L-amino acid oxidases, and C-type lectins [13,15,16]. Although these proteins have been extensively investigated for the development of antivenoms and therapeutic agents, certain proteins present in lower proportions have received little attention due to the challenges associated with their purification and characterization [17,18].

In this study, previously assembled transcriptomes of *Bothrops asper* and *Bothrops jararaca* [19], were used to identify and characterize low-abundance proteins that have been little explored in these species. The analysis focused on proteins such as arylsulfatase, CRISP (Cysteine-Rich Secretory Protein), SVMP-PIIIb metalloproteinases, von Willebrand factor type D, and dihydroorotate dehydrogenase, in order to predict their potential structural functions and their involvement in venom toxicity. In addition, proteins with limited representation in databases, such as botrocetin and basparin A, were searched for in the transcriptome of both *Bothrops* species to identify potential new isoforms.

Although these proteins are in low abundance, their analysis could reveal specialized functions that have gone unnoticed in studies focused on the major venom components. We hypothesize that these low-abundance proteins play key functional roles in modulating toxicity, target tissue specificity, or evading the prey’s immune system, and that these functions may be determined by the conservation of functional structural domains. To this end, we identified, described, and structurally modeled them using bioinformatics tools such as AlphaFold2. The predictions were compared with available crystallographic structures to predict their functions and assess their relevance to venom composition and toxicity. Since many of these proteins are difficult to isolate by conventional laboratory methods, we propose that a bioinformatics approach represents an effective way to characterize them initially, providing a solid foundation for future functional studies involving peptide synthesis or expression in cellular systems.

## 2. Results and Discussion

### 2.1. CRISP

Allergen 5 (Ag5) is a major protein component of the venom of hymenopterans such as wasps and ants belonging to the families Vespidae and Formicidae [20]. Previous studies have identified this low molecular weight peptide (~23 kDa) as biologically active and involved in reactions such as edema, urticaria, anaphylaxis, and allergic responses [21,22]. Ag5 belongs to the CAP superfamily, which includes cysteine-rich secretory proteins (CRISPs), allergen 5 (Ag5), and pathogenesis-related proteins (PR-1) [23,24,25]. In hymenopterans, the genus *Vespula* presents high levels of allergen 5, accounting for up to 23% of the venom [22,23,24,25]. On the other hand, CRISPs have been identified in the venoms of snakes, spiders, cone snails, and scorpions [26,27]. Studies in the genus *Bothrops* have shown that this protein constitutes between 0.2% and 1.6% of the total venom; however, the role of CRISPs during snake envenomation remains poorly understood [28,29,30]. Currently, CRISPs from the genus *Bothrops* listed in the UniProt and NCBI databases are reported in *Bothrops jararaca*, *Bothrops cotiara*, *Bothrops fonsecai*, *Bothrops moojeni*, and *Bothrops atrox*, although none of them have an available three-dimensional model.

To explore the presence of Ag5-like proteins in *Bothrops* venoms, we used the amino acid sequence of Ag5 from *Vespula germanica* as a reference to search for homologous sequences in transcriptomic data from *Bothrops asper* and *Bothrops jararaca*. The model generated with AlphaFold2 for the sequences identified in both *Bothrops* species revealed structural similarities with the Ag5 protein of *V. germanica* (Figure 1D). The multiple sequence alignment of *B. asper* and *B. jararaca* sequences, as previously reported by Espín-Angulo and Vela (2024) [19], confirmed the presence of conserved domains characteristic of the CAP superfamily, including CRISP and Ag5 (Appendix A).

At the structural level, the models display an α-β-α sandwich fold (Figure 1), a feature characteristic of the CAP superfamily. This protein fold is highly conserved across various organisms, both vertebrates and invertebrates [23,26]. In wasps such as *Vespula vulgaris*, the α-β-α sandwich fold has been identified in allergen 5, while in snakes such as *Trimeresurus stejnegeri* [31], *Trimeresurus flavoviridis* [32], and *Naja atra* [33], it has been found in the CRISP structure. Another conserved feature of the CAP superfamily is the N-terminal CAP domain and the C-terminal CRISP domain [23,26]. The presence of these domains and conserved sites in allergen 5 was confirmed by PFAM analysis for the *Bothrops asper* and *Bothrops jararaca* sequences (Appendix A).

Despite differences in the disulfide bridges and conserved residues among *Bothrops asper*, *Bothrops jararaca*, and *Vespula germanica* (Figure 1), the Comparative Root Mean Square Deviation (RMSD) analysis showed high structural similarity, with values of 0.813 Å for *B. asper* and 0.717 Å for *B. jararaca*. These results are promising, as the study by Suzuki et al. (2008) [34] reported RMSD values below 1 Å when comparing pseudechetoxin (*Pseudechis australis*) and pseudecin (*Pseudechis porphyriacus*), two CRISPs. While the RMSD threshold considered acceptable may vary depending on the protein, values below 2 Å generally indicate high structural similarity, whereas values above 3 Å may suggest significant structural differences. However, this also depends on the size of the protein and its evolutionary relationship [35,36].

Based on these findings, the models of *Bothrops asper* and *Bothrops jararaca* appear to be more closely related to CRISPs than to allergen 5, even though the reference sequence used for the search was Ag5 from *Vespula germanica*. To verify this hypothesis, the *Bothrops* models were compared with known crystal structures of snake venom CRISPs, including triflin [32,37], stecrispin [31], and natrin [33] (Figure 2), derived from *Trimeresurus flavoviridis*, *Trimeresurus stejnegeri*, and *Naja atra*, respectively.

The α-β-α fold and the N-terminal region of *Bothrops asper* and *Bothrops jararaca* show high structural similarity with crystallized CRISPs such as triflin, stecrispin, and natrin (Figure 2). However, a detailed analysis of the N-terminal sequences reveals some differences. According to sequence composition, N-terminal regions can be classified into two types: kaouthin-1 and kaouthin-2 [37]. Matsunaga and colleagues (2009) [37] constructed a phylogenetic tree of CRISPs from snake venom, classifying Elapidae within the kaouthin-2 group. In contrast, Colubridae and Viperidae do not fall into a specific category and may belong to either group. The N-terminal region of kaouthin-2 may lack 6 to 10 residues, which are absent in *Bothrops asper* and *Bothrops jararaca*, but are present in *Trimeresurus flavoviridis*, *Trimeresurus stejnegeri*, and *Naja atra*. These findings demonstrate that despite the existence of different CRISP isoforms, the α-β-α sandwich fold remains conserved, suggesting an evolutionary stability of this protein structure across distinct phylogenetic groups [37].

On the other hand, the absence of C-terminal in *Bothrops asper* and *Bothrops jararaca* (Figure 2) may be related to the structural flexibility of these regions. Crystallographic studies have reported that the C-terminal region is highly flexible, which can hinder its detection in structural modeling analyses [32]. These differences may contribute to the diversity in CRISP interactions with ion channels, potentially leading to functional implications, as C-terminal often plays a crucial role in ion binding. In Figure 3A, the C-terminal region of triflin (*Trimeresurus flavoviridis*) contains residues that interact with cadmium. In this context, the absence of the C-terminal region in *B. asper* and *B. jararaca* may alter their ion-binding affinity or modify their specificity for certain ion channels, ultimately affecting their toxic or physiological function. Additionally, studies on other CRISPs suggest that variability in the C-terminal could influence their ability to block calcium or potassium channels, which may explain the physiological differences observed among different species [34].

Among the conserved ion-binding residues are histidines (highlighted in green), which are preserved in *Bothrops asper* (H58–H107) and *Bothrops jararaca* (H55–H104) and align with those found in the *Trimeresurus flavoviridis* model (H60–H115) (Figure 3B,C). These residues have the potential to coordinate cadmium, which can be replaced by zinc ions. In this context, zinc not only stabilizes the protein structure but also modulates its functional activity [26,37]. It plays a key role in regulating CRISP interactions with ion channels. For instance, natrin from *Naja atra* is known to block calcium-activated potassium (BKca) channels and ryanodine receptors (RyR), which mediate calcium release [38].

Therefore, we suggest the possibility that the CRISPs identified in *Bothrops asper* and *Bothrops jararaca* could play a similar role, favoring the activation of inflammatory responses through zinc-mediated mechanisms. This effect could have implications on local inflammation after the bite, contributing to the alteration of the endothelium, which could enhance the tissue damage observed in bothropypic envenomation [26,34,38].

### 2.2. Von Willebrand Factor Type D

Apolipophorin is a conserved protein in both invertebrates and vertebrates due to its crucial role in lipid transport [39,40]. This protein shares similarities with vitellogenin from both vertebrates and invertebrates, human apolipoprotein B, and von Willebrand factor type D [39,41,42,43]. During the search for apolipophorin in the transcriptome of *Bothrops asper* and *Bothrops jararaca*, no matching sequences were identified. However, in *B. jararaca*, a contig with similarity to von Willebrand factor type D (vWFD) was found. This sequence was used as a reference for the search for vWFD in *B. asper*. In the multiple alignment (Appendix A), a high conservation of the vWFD domain is observed in *B. asper* and *B. jararaca*, identified with PFAM.

It is important to note that, while the apolipophorin from *Odontomachus monticola* (trap-jaw ant) is a large protein (3375 amino acids), the analysis focused exclusively on the vWFD domain, which spans amino acids 2799 to 2968 (169 amino acids). In the case of *Bothrops asper*, the model was much more restricted to beta sheets (Figure 4A), while for *Bothrops jararaca*, the model showed an orientation similar to the vWFD domain of *O. monticola* apolipophorin (model generated with Swiss-Model) (Figure 4D).

Following this discovery, a more detailed analysis of the presence of vWFD in *Bothrops asper* and *Bothrops jararaca* was conducted, utilizing the crystallized structure of human vWFD (PDB 6N29). By performing tblastn, vWFD sequences were identified within the venom gland transcriptomes of both *Bothrops* species. An interesting observation was that the contigs identified based on the human vWFD were distinct from those identified using the apolipophorin of *Odontomachus monticola*. These newly identified sequences allowed for the generation of more complete models, which were then compared to crystallized structures (Figure 5).

PFAM analysis of these newly identified sequences from *B. asper* and *B. jararaca* revealed the presence of vWFD, C8, and TIL domains (Appendix A, Appendix A). However, it is noteworthy that the TIL domain was only observed in the *B. asper* model (Appendix A).

The analyzed models show structural similarities to the reference model, with RMSD values of 1.456 Å for *Bothrops asper* and 1.050 Å for *Bothrops jararaca*. However, despite this global similarity, the *Bothrops* models exhibit structural differences. In general, the vWFD domain features a central fold composed of 12 β-strands [44], however, in the reference model, only 8 β-strands were identified, 7 β-strands in *Bothrops jararaca*, and 4 β-strands in *Bothrops asper* (Figure 5).

Each *Bothrops* model presents distinct structural characteristics. The *Bothrops asper* model, for instance, is the only one that retains the TIL domain (Figure 6A), which shows similarity to the D3 region of human vWF, a crucial part for structural stabilization and interaction with factor VIII (FVIII) [44,45]. A key element of this region is the Cys1153-Cys1165 disulfide bond, whose conservation helps stabilize the domain and its ability to interact with coagulation factor VIII [44].

The model generated with AlphaFold indicates that *B. asper* retains two cysteine residues in the Cys134-Cys147 region, separated by 1.634 Å, allowing for the formation of a disulfide bond. The conservation of this structure suggests potential stability in the TIL domain architecture, although identical functionality to human vWF cannot be directly assumed, as some residues in the generated model differ from the reference model (Appendix A). Despite the structural differences, the identification of a sequence with similarity to the vWFD domain in *B. asper* suggests a potential role in the interactions with coagulation factors such as FVIII and vWF. This is consistent with the known protein diversity of *Bothrops* venom, where proteins with similar functions have been previously described. Toxins such as bitiscetin and bothrocetin have been reported in certain species of this genus, capable of modulating prey coagulation by interacting with these factors and enhancing the hemorrhagic effects of the venom [46,47,48,49,50,51,52].

On the other hand, the C8 domain of *Bothrops jararaca* and human vWFD maintain the same structure composed of four α-helices [44]. However, there are differences in the disulfide bonds, as in *B. jararaca*, Cys1190 is absent (Appendix A). This change results in the formation of only four disulfide bonds instead of the expected five. Similar alterations in cysteine numbers have been described in humans, where mutations that eliminate or add cysteines can impact disulfide bond formation and vWFD function [44,45].

The observed differences in the vWFDs of *Bothrops asper* and *Bothrops jararaca* are particularly interesting as they highlight how snakes exhibit their own adaptations. Although the effects in mammals remain uncertain, we propose that modifications in the *B. jararaca* vWFD could affect its binding to mammalian FVIII, interfering with its multimerization and increasing the risk of hemorrhage [53]. Similar effects have been observed in other snake venoms, where C-type lectins and type III metalloproteinases inhibit the function of mammalian vWF, impairing platelet adhesion and promoting bleeding [54].

### 2.3. Arylsulfatase

The sulfatase family is conserved across both eukaryotes and prokaryotes and includes various types (A–K), with arylsulfatase A (ARSA) and arylsulfatase B (ARSB) being the most extensively studied in mammals [55,56,57]. ARSA and ARSB play crucial roles in the degradation of sulfatides and glycosaminoglycans (GAGs), a process essential for the recycling of molecular components and the maintenance of proper extracellular matrix function [56,58]. GAGs are abundant in tissues such as tendons, joints, and the vascular endothelium, and they also participate in the regulation of blood coagulation and inflammatory responses [59,60,61,62].

In the mollusk *Charonia tritonis*, Bose et al. (2017) [63], identified ARSB and suggested that its ability to degrade GAGs could be linked to digestive processes. In snakes, the presence of this protein has been reported only in *Naja nigricollis* (the black-necked spitting cobra), where it has been hypothesized that its action may be associated with the degradation of GAGs in structures such as tendons, connective tissue, and ligaments [64].

Within the Viperidae family, arylsulfatase sequences are recorded in the NCBI/UniProt databases for *Crotalus tigris*, *Protobothrops mucrosquamatus*, and *Crotalus adamanteus*; however, all derive from genomic annotations. The most recent reports of this enzyme’s presence in venom glands have emerged from studies on the ectoparasitoid wasp *Torymus sinensis* [65], the scorpion *Diplocentrus whitei* [66], and the centipede *Theatops posticus* [67].

Furthermore, arylsulfatase has been identified in the venom gland transcriptomes of *Bothrops asper* and *Bothrops jararaca*, with multiple sequence alignment confirming the presence of both sulfatase and arylsulfatase domains [19]. Structural models generated using AlphaFold2 based on the sequences identified in *B. asper* and *B. jararaca* revealed similarities to the arylsulfatase of *Ophiophagus hannah* (Figure 7).

The models show a high structural similarity to *Ophiophagus hannah*, although the *Bothrops jararaca* model is smaller, with RMSD values of 1.706 Å, and *Bothrops asper* of 1.180 Å. These differences may be related to variations in the number of β-sheets, with 4 β-sheets in *B. asper*, 3 β-sheets in *B. jararaca*, and 12 β-sheets in *O. hannah*.

Despite these variations, the sulfatase domain is conserved in all three species, maintaining the same key residues (C/S-X-P-X-R-X_4_-TG) and the characteristic α-helix (Appendix A) [55]. In both prokaryotes and eukaryotes, this motif is highly conserved, as it is essential for the degradation of glycosaminoglycans (GAGs) [55,68,69,70].

Its presence in the venom gland transcriptomes of *Bothrops asper* and *Bothrops jararaca* suggests that, despite differences in size, the domain organization and essential structural elements of the enzyme are preserved. Therefore, arylsulfatases in both species may play a similar role in GAGs degradation in prey tissues, potentially contributing to tissue damage, ligament and tendon degradation, or even digestion [63,64]. Similar effects have been proposed for arylsulfatases found in the venom of *Naja nigricollis*, *Diplocentrus whitei*, and *Theatops posticus* [64,66,67,68].

In humans, it has been demonstrated that arylsulfatase B (ARSB) requires a metal ion to stabilize the formation of the sulfate ester, which can bind to biomolecules such as glycosaminoglycans [55,71]. When comparing the human ARSB model (PDB 1FSU) with those of *Bothrops*, only the *Bothrops asper* model retains the same residues involved in calcium binding [71] (Appendix A, Appendix A) (Figure 8).

The calcium ion in *Bothrops asper* and human ARSB is coordinated by the same residues: Asp54, Cys91, Arg95, Lys145, His147, His242, Asp300, Asn301, and Lys318 (Appendix A). In the human enzyme, Cys91 undergoes a post-translational modification to form Cα-formylglycine (FGly), a critical residue involved in the degradation and recycling of sulfated biomolecules, as it contributes to calcium stabilization within the active site [55,71]. In contrast, the *B. asper* model reveals the presence of an unmodified Cys91, with no evidence of FGly formation, suggesting potential differences in the enzyme’s activation mechanism or in the regulatory factors governing the cysteine-to-FGly conversion. Nevertheless, the conservation of the calcium-coordinating residues in both species supports the presence of a highly conserved catalytic mechanism throughout evolution [55,71,72].

The conservation of Cys91 in the arylsulfatase of *Bothrops asper* may indicate a role in the degradation of sulfated biomolecules such as glycosaminoglycans, tendons, or ligaments, a process demonstrated in the venoms of other species, as discussed. Since Cys91 has been implicated in the stabilization of sulfate esters, its presence underscores its potential functional relevance in enzymatic activity and its possible involvement in the toxic or digestive processes of *B. asper*. These findings suggest that the identified sequences may contribute to the venom’s toxicity mechanisms by promoting interactions with specific biological targets and enhancing the effects of other toxins.

### 2.4. Botrocetin

Botrocetin is a well-characterized venom protein from *Bothrops jararaca*, known for its interaction with the mammalian von Willebrand factor (vWF) [46,47,48,49,50]. During envenomation, botrocetin induces hemostatic disturbances, including the consumption of clotting factors V and VIII, inflammation, and thrombocytopenia [46,73]. Thrombocytopenia, defined as a reduction in circulating platelet count, impairs proper coagulation and facilitates hemorrhagic events [74,75]. These effects result from botrocetin’s interaction with vWF, promoting platelet agglutination, which ultimately reduces platelet availability and triggers bleeding [47].

Within the Viperidae family of toxins, botrocetin belongs to the C-type lectin-like proteins, a group of non-enzymatic molecules highly abundant in *Bothrops* venoms. These proteins modulate platelet function, disrupt coagulation cascades, and influence endothelial and immune cell activity [46,47,76].

Botrocetin is a disulfide-linked heterodimer composed of two homologous chains, A and B [47,77]. Chain A exhibits strong similarity to canonical C-type lectin-related proteins, whereas the B chain is unique to botrocetin and confers its specific affinity for vWF [77]. For this reason, our study focuses on the B chain, aiming to identify similarities between the *B. jararaca* botrocetin and a putative homolog identified in *Bothrops asper* from venom gland transcriptomic data (Figure 9). This sequence was retrieved through a tblastn analysis using the *B. jararaca* botrocetin as the query.

The typical botrocetin structure consists of two α-helices and five β-strands, which are also present in the *Bothrops asper* and *Bothrops jararaca* models, despite some variations in specific residues (Appendix A). Despite these differences, the models retain the arrangement of disulfide bridges, which are essential for protein stability (Figure 10). According to the study by Sen et al. (2000) [77], the magnesium-binding site in chain B is located near the concave surface of botrocetin, a critical region for its interaction with the A1 domain of vWF. In the crystallographic model of *B. jararaca* (1FVU), the residues coordinating the Mg^2+^ ion are Ser441 (S), Gln443 (Q), Glu447 (E), and Glu522 (E) (Figure 10C). These residues are conserved in *B. asper* (Figure 10A); however, in *B. jararaca*, a substitution of Glu with Lys is observed.

Since chain B is the primary region responsible for vWF binding, this variability could influence differences in affinity or specificity between species [77]. It has been observed that in other proteins, such as the histidine kinase EnvZ, the substitution of a residue near the cation-binding site can allow the protein to function without the need for the metal ion [78]. Similarly, the presence of Lys instead of Glu at the Mg^2+^ binding site in *Bothrops jararaca* could indicate a reduced dependence on Mg^2+^ to stabilize the botrocetin structure, enabling it to maintain its function [77].

As documented, the three-dimensional models exhibit a remarkable similarity between the reference botrocetin model and the model generated for *Bothrops asper* (Figure 9 and Figure 10). Although botrocetin is a protein characteristic of the *B. jararaca* venom, the analyzed *B. asper* sequence could correspond to aspercetin, a hypothesis supported by the structural similarity between both proteins. Both botrocetin and aspercetin are heterodimers of the C-type lectin family and share the ability to interact with vWF [46,79,80,81].

UniProt currently provides only two models for aspercetin, corresponding to chains A and B, each comprising 62 amino acids. However, chain B analyzed in this study is 123 amino acids long, leading to significant structural differences. Compared to the UniProt aspercetin model, the *Bothrops asper* sequence exhibits an extended length, a greater number of disulfide bonds, and improved conservation of the magnesium-binding sites (Figure 11). These features are critical for interacting with von Willebrand factor (vWF) and the platelet receptor GPIb, facilitating rapid platelet depletion and prolonged hemorrhage [81,82]. The absence of these key elements in the UniProt model suggests that the model generated in this study more accurately and comprehensively reflects the structure of *B. asper* aspercetin compared to the models available in databases.

### 2.5. Dihydroorotate Dehydrogenase

Dihydroorotate dehydrogenase (DHODH) is a key enzyme in pyrimidine synthesis, catalyzing the oxidation of dihydroorotate to orotate a process intricately linked to energy metabolism and cell proliferation [83,84]. DHODH is classified into two main types: class 1, found in most bacteria and some protists, and class 2, which is characteristic of eukaryotes [85,86,87]. In humans, DHODH belongs to class 2 and is localized in the inner mitochondrial membrane, predominantly expressed in metabolically active tissues such as the liver, immune cells, and lymphatic tissue [86,88,89,90,91].

In this study, conserved domains and active sites of DHODH were identified in the venom gland transcriptomes of *Bothrops asper* and *Bothrops jararaca* (Appendix A). It is well established that the venom glands of crotalids and viperids consist of both a main gland and an accessory gland [92,93,94]. The main gland is responsible for toxin synthesis and is composed of cells rich in mitochondria [93]. In *Crotalus viridis oreganus* (western rattlesnake), for example, it has been shown that following venom extraction, the main gland increases the production of both toxins and mitochondrial components [93]. These cells not only compensate for the high metabolic demand of toxin production but are also implicated in venom stabilization [95].

The structure of DHODH features two characteristic domains: a small N-terminal domain composed of two alpha helices (α1 and α2), and a large C-terminal domain formed by eight beta sheets (β1–β8) surrounded by eight alpha helices (α3, α4, α6–α10, and α12) (Appendix A) [84].

The N-terminal domain facilitates binding to the inner mitochondrial membrane, allowing electron transfer [83,84]. Additionally, it interacts with highly conserved residues such as Lys225, Ser215, Asn145, and Asn284, which are essential for the reduction in flavin mononucleotide (FMN) to FMNH2 [83,84]. This reduction is critical for the C-terminal domain to transfer electrons in the conversion of dihydroorotate to orotate [83,84,96]. These residues are conserved in *Bothrops asper* and *Bothrops jararaca* (Appendix A); however, *B. jararaca* lacks the N-terminal domain (Figure 12B). Despite this absence, Copeland et al. (1995) [97] noted that the lack of N-terminal residues does not hinder interaction with the inner mitochondrial membrane. The loss of amino acids in this region has been reported in human DHODH [84] and in *Escherichia coli* [97], without affecting its catalytic function. This suggests that DHODH from *B. jararaca* could employ alternative mechanisms to associate with the mitochondrial membrane.

The conversion of dihydroorotate to orotate is a crucial process for pyrimidine synthesis, occurring at the active site of DHODH, located within the large α/β domain [84,87]. This structure is highly conserved in *Bothrops asper* and *Bothrops jararaca* (Figure 12), with key active site residues identified in both models (Ser215, Tyr356, Tyr147, Asn145, Gly148, Asn212, Asn284, Gly306, Gly335, Gly97, Gln225, Lys100, Ser120, and Thr357) (Appendix A). The interaction of dihydroorotate with FMN is mediated by Lys100 and Lys225, which form hydrogen bonds, stabilizing its position in the active site for oxidation [84]. The oxidation of dihydroorotate is driven by the action of several residues. Ser215 acts as a catalytic base, directly participating in the electron transfer from dihydroorotate to FMN [84]. Meanwhile, Tyr356 and Tyr147 enhance the affinity between FMN and dihydroorotate, promoting their interaction and facilitating oxidation [84]. As a result, dihydroorotate donates its electrons, leading to the formation of orotate [87].

This oxidation process is crucial for nucleotide synthesis, which is vital for cell proliferation and DNA replication [87,96,98]. The presence of these residues in the DHODH models of *Bothrops asper* and *Bothrops jararaca* highlights the importance of this reaction in cellular physiology. Moreover, their conservation suggests preserved functionality in pyrimidine synthesis, which could have implications for the metabolic regulation of the venom gland in these species.

### 2.6. Basparin

Basparin is a type III snake venom metalloproteinase (SVMP-III) isolated from the venom of *Bothrops asper*, characterized as a type A prothrombin activator that functions independently of calcium and coagulation cofactors V and X [99,100,101]. Similar proteins have been identified in other species, including ecarin from *Echis carinatus* [100], jararhagin from *Bothrops jararaca* [102], and catrocollastatin from *Crotalus atrox* [103], some of which have resolved three-dimensional structures. SVMPs are generally classified into three main structural types, with the P-III class being the most complex due to the presence of a metalloproteinase domain, a disintegrin-like, and a cysteine-rich domain. These proteins are known to induce hemorrhage and activate prothrombin, contributing to thrombosis [101,104,105,106].

Although most P-III SVMPs share these conserved domains, post-translational modifications have led to the emergence of five distinct subclasses: P-IIIa to P-IIIe [107,108]. P-IIIa SVMPs retain all three domains intact, with representative members including AaHIV from *Deinagkistrodon acutus* [109] and ohagin from *Ophiophagus hannah* [110]. In contrast, P-IIIb SVMPs undergo a proteolytic separation of the metalloproteinase domain from the disintegrin-like and cysteine-rich domains [104]. This subclass includes proteins such as alternagin from *Bothrops alternatus* [111] and VAP2B from *Crotalus atrox* [112], both of which inhibit collagen-induced platelet aggregation [111,112].

P-IIIc SVMPs can form homodimeric structures, which enhance their proteolytic activity and structural stability [104,107]. Notable examples include VAP1 from *Crotalus atrox* [113] and VaH3 from *Vipera ammodytes ammodytes* [114], both of which exhibit increased proteolytic efficiency against factor X, collagen (types I and IV), and fibronectin [113,114]. The P-IIId subclass is structurally similar to P-IIIa members but differs by forming multimeric complexes with C-type lectin-like proteins [104,107]. One representative is RVV-X from *Daboia russelii* [115,116], whose three-dimensional conformation enables interactions with membrane receptors and coagulation factors, resulting in prey coagulopathies [104,115].

Finally, the P-IIIe subclass stands apart from the others, as it completely lacks the metalloproteinase domain. To date, the only known member of this group is Vaa-MPIII-3 from *Vipera ammodytes ammodytes* [117]. These variations highlight the relevance of exploring structure–function relationships within SVMPs, as they display a broad range of structural and functional diversity. For this reason, Basparin was used as a reference to search for it in the transcriptome of the venom gland of *Bothrops asper*, leading to the identification of a matching sequence, here referred to as Ba_1.

In UniProt and NCBI databases, only a single entry is available for basparin from *Bothrops asper* (P84035). However, when this protein was queried against the *B. asper* venom gland transcriptome, a sequence exhibiting notable differences in residues was identified (Appendix A, Appendix A), resulting in a predicted structural model with a distinct orientation compared to the reference (Figure 13). Structurally, both models share the presence of an α-helix (Figure 13C), despite the variation in residues (Appendix A). Additionally, both sequences conserve the functional HEXXH motif, although located in different secondary structures: in basparin, it is embedded within a β-sheet, whereas in the transcript-derived sequence (Ba_1), it is situated in an α-helix (Appendix A).

Several SVMP-III proteins described in the literature, such as AaHIV from *Agkistrodon acutus* [109], bothropasin from *Bothrops jararaca* [118], VAP1 from *Crotalus atrox* [113], and VaH3 from *Vipera ammodytes ammodytes* [114], exhibit a conserved motif similar to those found in our models. However, this motif is typically longer (HEXXHXXGXXH), located within an α-helix, and functionally associated with zinc coordination. The conservation of this motif is critical, as the spatial arrangement of histidine and glutamate enables zinc binding, with the glutamate residue coordinating a water molecule that acts as a catalytic base during enzymatic activity [119,120,121].

Based on these findings, we propose that the transcript-derived sequence (Ba_1) shows greater similarity to hemorrhagic P-III SVMPs involved in zinc and calcium binding, suggesting it is unlikely to be a basparin isoform. To test this hypothesis, bothropasin from *Bothrops jararaca* was used as a reference in the analysis of the *Bothrops asper* transcriptome. This approach yielded a sequence similar to Ba_1 (Figure 14A), but was considerably larger, and we designated this sequence as “B_asper_bothropasin_AF” to avoid confusion. PFAM domain analysis revealed the presence of a metalloproteinase domain, an ADAM domain, and a canonical zinc-binding active site (HEXXHXXGXXH) within this sequence (Appendix A, Appendix A).

Bothropasin from *Bothrops jararaca* is a zinc-dependent, hemorrhagic metalloproteinase belonging to the SVMP-PIII class [118]. This monomeric enzyme comprises three domains: a metalloproteinase domain (M), a disintegrin-like domain (D), and a cysteine-rich domain (C). Among these, only the M domain is shared with the predicted model “B_asper_bothropasin_AF” (Appendix A). The zinc-binding motif (HEXXHXXGXXH) is conserved within an alpha-helix of the M domain, coordinated by the histidine residues His145, His149, and His155 (Appendix A) [118]. Within this domain, six cysteine residues, Cys120–Cys200, Cys160–Cys184, and Cys162–Cys167, are conserved, forming three disulfide bridges in both the bothropasin and B_asper_bothropasin_AF models (Appendix A). Additionally, the model retains two of the three calcium-binding sites. The first, located within the M domain, involves residues Glu12, Asn203, Asp96, and Cys200, which have been previously reported in other SVMPs and are thought to contribute to structural stability alongside disulfide bonds [120,122]. The second calcium-binding site is located within the D domain and is formed by Val215, Gln218, Leu220, Glu222, and Glu225. This site lies in proximity to the first ECD motif (disintegrin loop), which is highly conserved in SVMP-PIII proteins and composed of Glu226, Cys227, and Asp228 (Appendix A) [118].

The comparison between the B_asper_bothropasin_AF model and the metalloproteinases bothrasperin (Q072L5) and BaP1 (P83512) from *Bothrops asper* revealed structural similarity limited to the M (metalloproteinase) domain (Appendix A; Figure 15). This result was expected, as bothrasperin and BaP1 belong to the P-II and P-I classes of SVMPs, respectively [123,124]. Having already compared our model with basparin A (P84035), bothrasperin (Q072L5), and BaP1 (P83512), which are well-characterized metalloproteinases from *B. asper* available in NCBI, we propose that the sequence identified in this study corresponds to a P-IIIb subclass SVMP. We ruled out the possibility of it being a P-II or P-I SVMP based on the absence of disintegrin-binding regions found in Q072L5 and P83512, as well as the presence of the calcium-binding site in the D domain, which is unique to our model (Appendix A).

A representative of the SVMP-P-IIIb subclass is VAP2B from *Crotalus atrox* (Q90282), whose model aligns closely with ours (Figure 15B), as previously observed with bothropasin. This further supports the hypothesis that the identified sequence represents a novel P-IIIb SVMP in *Bothrops asper*. One of the defining features of this subclass is its post-translational processing, where the metalloproteinase (M) domain is cleaved from the disintegrin-like (D) and cysteine-rich (C) domains [104,125,126]. It is likely that our modeled sequence has undergone this process, retaining only the M domain. However, the disintegrin-binding region (ISPPVCGNELLEV) remains conserved (Appendix A), as observed in proteins such as bothropasin and VAP2B [118,127].

Together, the structural and comparative analyses suggest that the identified sequence represents a P-IIIb SVMP from *Bothrops asper*, with key features conserved among metalloproteinases of this subclass. The preservation of the M domain, calcium-binding capacity, and the disintegrin-binding region all strengthen this hypothesis. Additionally, the high degree of similarity with *Crotalus atrox* VAP2B and bothropasin supports the proposition that this sequence may represent a novel P-IIIb SVMP in *B. asper*.

### 2.7. Model Quality Analysis

Most of the predicted models exhibit structurally stable conformations; however, some show low Verify3D scores, despite the vast majority of residues being in favored positions in the Ramachandran plots (Table 1). This result is not alarming, as the low Verify3D scores may be due to errors in the modeling of flexible regions, making structural analysis difficult by the program. A representative example is the model of the protein basparin, whose Verify3D score was low (1.79%), but it is based on a reference sequence whose own structure has a moderate Verify3D score (55.56%) with poorly scored flexible regions, suggesting that this limitation is related to the intrinsic nature of the protein rather than to modeling errors.

On the other hand, the ERRAT analysis indicates that values above 80% correspond to high-quality models; however, previous studies have determined that values above 50% may still be acceptable within the structural validation criteria [128,129]. Considering all these validation parameters, the predicted models for *Bothrops asper* and *Bothrops jararaca* can be considered structurally reliable. Detailed information on each model can be found in Appendix A.

## 3. Conclusions

The results obtained highlight the importance of analyzing low-abundance proteins in the venom gland transcriptomes of *Bothrops asper* and *Bothrops jararaca*, revealing key findings that offer new insights into their potential roles in envenomation. The use of computational biology tools such as AlphaFold enabled the generation of high-quality structural models, which were compared with crystallized proteins available in public databases. These technologies are essential for characterizing low-abundance proteins that cannot be isolated through conventional experimental approaches.

In this study, structural models of proteins potentially relevant to venom toxicity were identified and validated, including arylsulfatases, CRISPs, von Willebrand factor-related proteins, and dihydroorotate dehydrogenase. In addition, potential new isoforms of aspercetin and basparin were characterized. These models were validated and found to be structurally reliable, providing a solid foundation for future functional and structural studies aimed at better understanding their roles in venom mechanisms of action.

## 4. Materials and Methods

To analyze protein abundance, transcript expression was quantified using TPM (transcripts per million) values obtained with Salmon v1.10.3. The information is summarized in Appendix A. Appendix A presents total TPM, mapped reads, and percentage relative expression. Proteins such as von Willebrand factor type D, arylsulfatase, and dihydroorotate dehydrogenase were detected with low but consistent expression.

Details by contig are included in the Appendix A TPM_BA for *Bothrops asper* and TPM_BJ for *Bothrops jararaca* in Appendix A. We recognize that, due to their low abundance, these transcripts should be interpreted with caution. However, their expression in multiple contigs, as evidenced in the detailed tables by protein, provides further support for their genuine presence in the venom gland transcriptome.

### 4.1. Data Collection

Protein sequences obtained from UniProt of apolipophorin, allergen 5, arylsulfatase, botrocetin, dihydropyrimidine dehydrogenase, aspercetin, and basparin A from different species were used as references (Table 2). A TBLASN search of the assembled transcriptome of the venom glands of *Bothrops asper* and *Bothrops jararaca* (Zenodo: https://zenodo.org/records/14862557 (accessed on 20 March 2025) identified sequences matching the reference proteins. This search was used to identify potential new toxic proteins in *Bothrops* species.

### 4.2. Identification of Domains and Active Sites

The prediction of protein families, motifs, domains, and active sites was performed using the Pfam protein database from InterPro. Sequences used in this step were those with e-values close to zero, in order to retrieve the most comprehensive information from the newly identified proteins. Pfam results were aligned using MUSCLE v.5.3 to assess sequence similarity. All materials, including sequences, alignments, and Pfam results for each protein, are provided in the Appendix A.

### 4.3. Protein Modeling

To predict the tertiary structures of the proteins, we used the AlphaFold2 platform via Google Colab. This tool leverages deep neural networks and advanced machine learning algorithms to analyze amino acid sequences and accurately infer their three-dimensional conformations [130,131]. Unlike traditional modeling tools such as MODELLER and SWISS-MODEL, AlphaFold2 does not rely on structural templates. Instead, it utilizes multiple sequence alignments (MSAs), geometric attention mechanisms, and optimization through the AMBER force field to refine structural predictions. This approach is particularly advantageous for modeling proteins that lack experimentally resolved structures. The resulting models were visualized using CHIMERA v1.17.3 and compared to crystallized protein structures retrieved from the Protein Data Bank (Table 3).

### 4.4. Model Validation

Model validation was performed using the software tools PROCHECK [132], ERRAT [133], and Verify3D [134], all accessible through the SAVES v6.1 server provided by the Institute for Genomics and Proteomics (IGP). Available at: https://saves.mbi.ucla.edu/ (accessed on 20 March 2025).

## Figures and Tables

**Figure 1 toxins-17-00262-f001:**
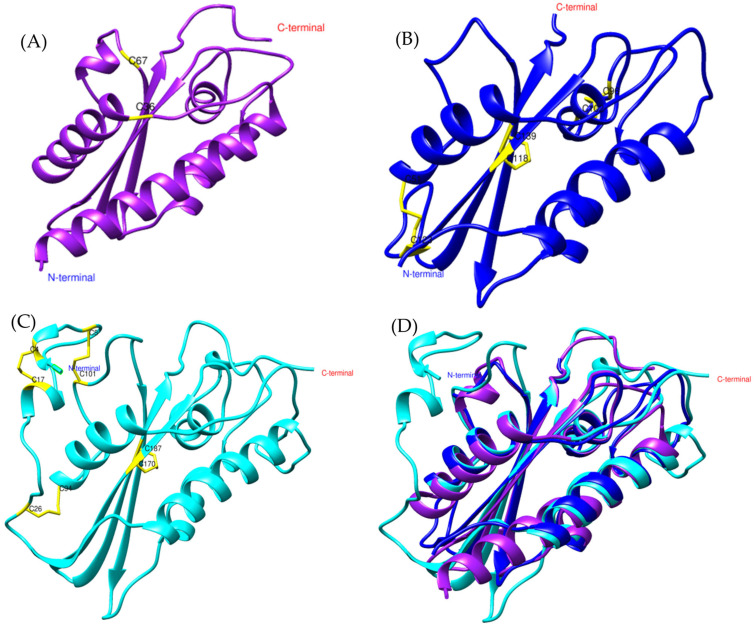
General structure of allergen 5 predicted by AlphaFold2. (**A**) In purple, *Bothrops asper*, (**B**) blue, *Bothrops jararaca*, and (**C**) cyan, *Vespula germanica* [UniProt P35784] (German wasp). The regions highlighted in yellow indicate cysteine residues (CYS) involved in disulfide bond formation. (**D**) Structural comparison of the three models.

**Figure 2 toxins-17-00262-f002:**
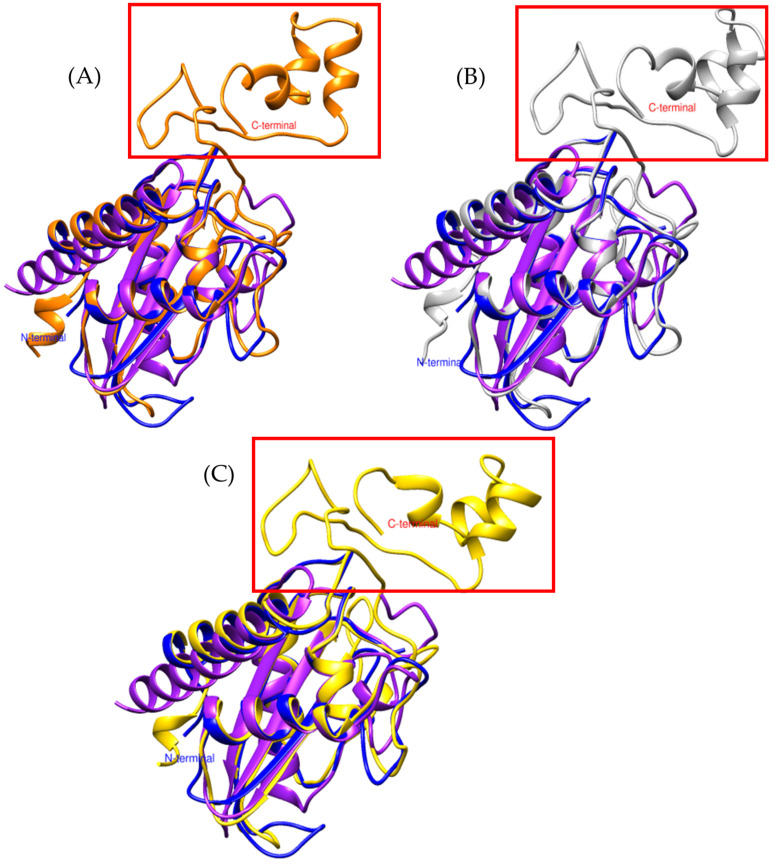
Comparison of CRISP models. (**A**) In orange, triflina [PDB 1WVR] (*Trimeresurus flavoviridis*), (**B**) white, stecrispina [PDB 1RC9] (*Trimeresurus stejnegeri*) and (**C**) yellow, natrin [PDB 1XTA] (*Naja atra*). The purple model corresponds to *Bothrops asper* and the blue to *Bothrops jararaca*.

**Figure 3 toxins-17-00262-f003:**
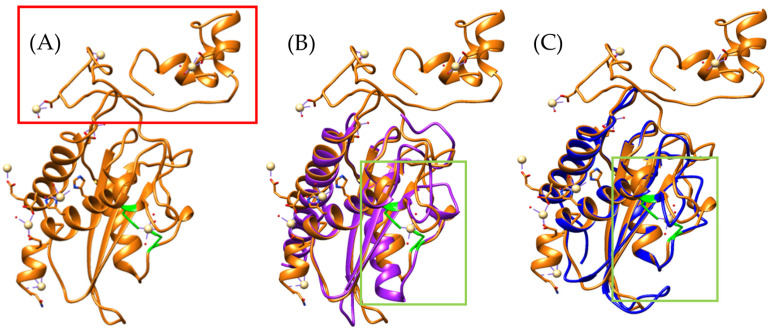
Identification of residues with cadmium-binding affinity. (**A**) Orange, Triflin [PDB 1WVR] (*Trimeresurus flavoviridis*), (**B**) superposition of the *Bothrops asper* model with *T. flavoviridis*, and (**C**) superposition of the *Bothrops jararaca* model with *T. flavoviridis*. Regions highlighted in green correspond to residues with potential cadmium-binding capacity. The red box indicates the C-terminal region. The *Bothrops asper* and *Bothrops jararaca* models are colored purple and blue, respectively.

**Figure 4 toxins-17-00262-f004:**
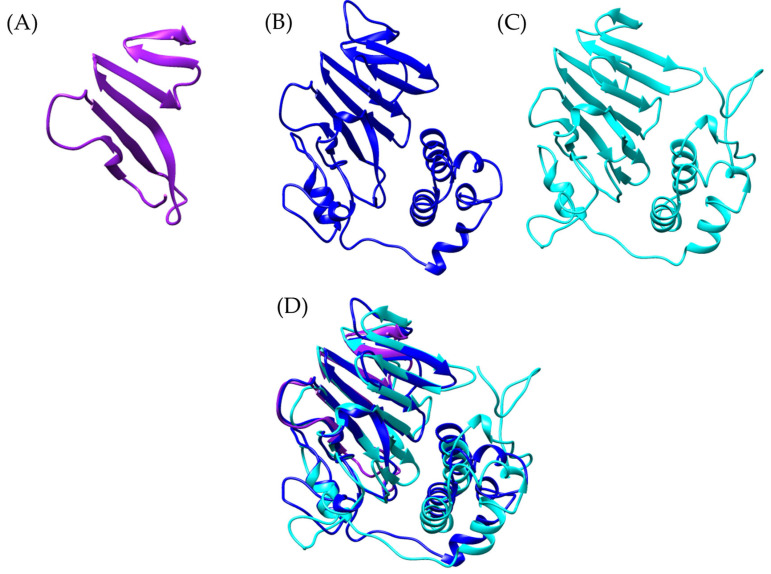
Structural comparison of von Willebrand factor type D models. (**A**) In purple *Bothrops asper*, (**B**) blue, *Bothrops jararaca*, and (**C**) cyan, *Odontomachus monticola* (trap-jaw ant). (**D**) Structural comparison of the three models.

**Figure 5 toxins-17-00262-f005:**
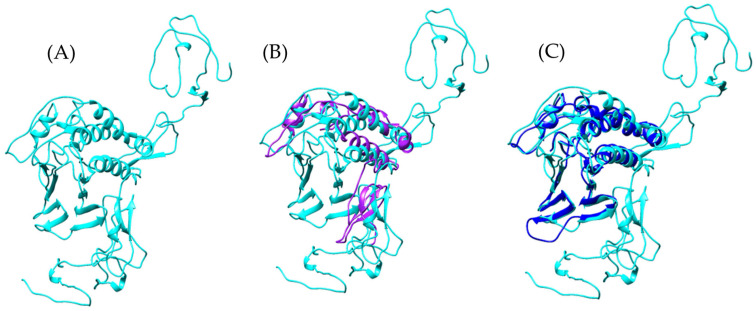
Comparison of vWFD models. (**A**) In cyan reference model of vWFD [PDB 6N29] (*Homo sapiens*), (**B**) superimposition of the model of *Bothrops asper* with reference, (**C**) superimposition of the model of *Bothrops jararaca* with reference. The *Bothrops asper* and *Bothrops jararaca* models are colored purple and blue, respectively.

**Figure 6 toxins-17-00262-f006:**
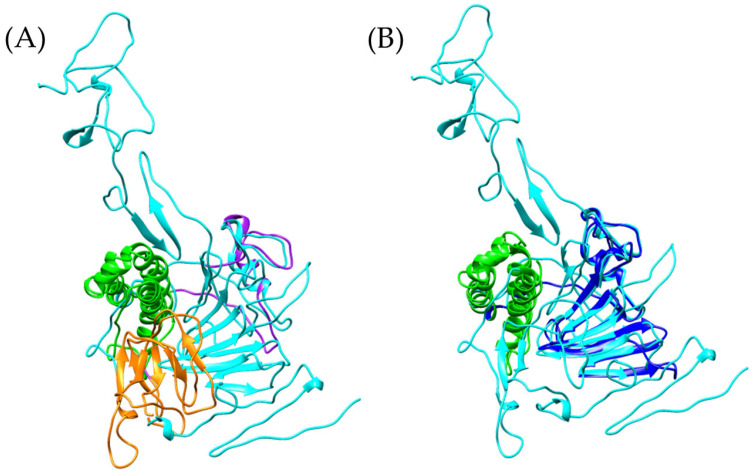
Comparison of vWFD models. The reference vWFD model from *Homo sapiens* [PDB 6N29] is highlighted in cyan. (**A**) Superposition of the reference model with *Bothrops asper*, (**B**) superposition of *Bothrops jararaca* with the reference model. The C8 domain is shown in green, and the TIL domain in orange. The *Bothrops asper* and *Bothrops jararaca* models are colored purple and blue, respectively.

**Figure 7 toxins-17-00262-f007:**
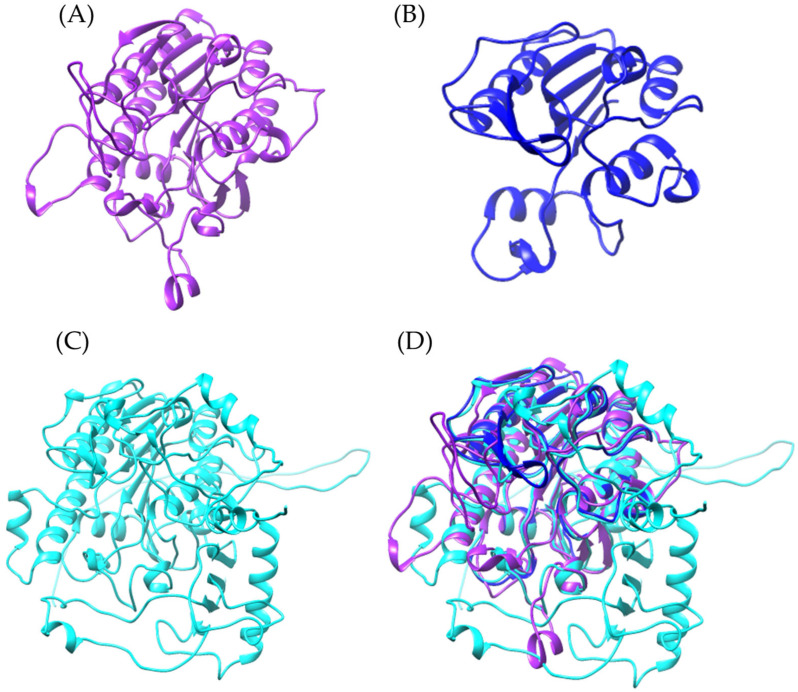
Comparison of arylsulfatase models. (**A**) *Bothrops asper*, (**B**) *Bothrops jararaca*, and (**C**) *Ophiophagus hannah* [UniProt V8PBU1]. (**D**) Structural superposition of the three models. The *Bothrops asper* and *Bothrops jararaca* models are colored purple and blue, respectively.

**Figure 8 toxins-17-00262-f008:**
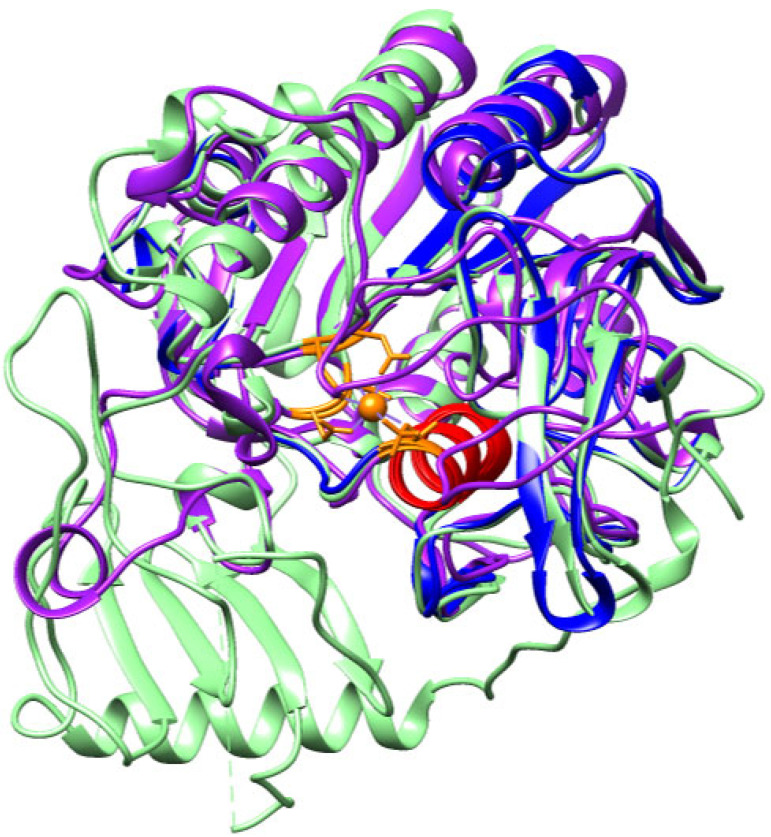
Structural alignment of the arylsulfatase models from *Bothrops asper* (purple), *Bothrops jararaca* (blue) and *Homo sapiens* (green). The orange region highlights the calcium-binding residues identified in the human model (PDB 1FSU), while the conserved sulfatase motif in the α-helix is shown in red.

**Figure 9 toxins-17-00262-f009:**
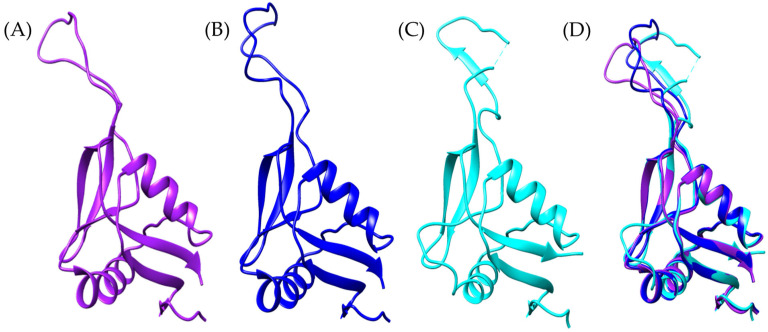
Botrocetin-like structural models. (**A**) *Bothrops asper* (purple), (**B**) *Bothrops jararaca* (blue) (transcriptome-derived), (**C**) crystallographic structure of *B. jararaca* botrocetin (cyan) [PDB: 1FVU], and (**D**) structural superposition of the three models.

**Figure 10 toxins-17-00262-f010:**
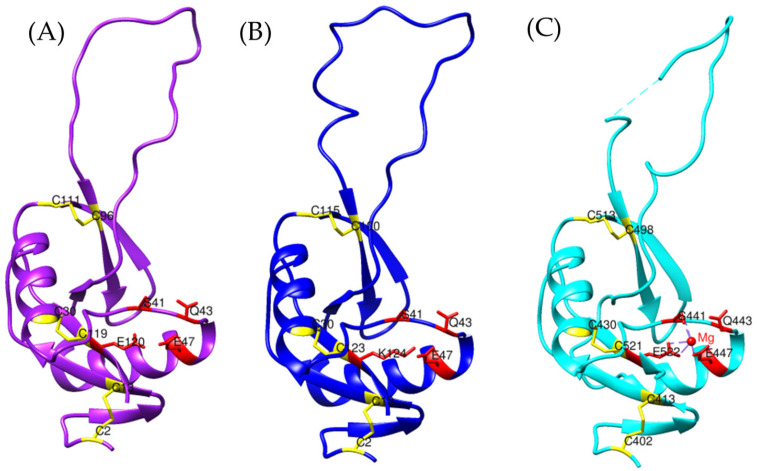
Magnesium-binding sites in botrocetin-like models. (**A**) *Bothrops asper* (purple), (**B**) *Bothrops jararaca* (blue), and (**C**) crystallographic structure of *B. jararaca* botrocetin (cyan) [PDB: 1FVU]. Disulfide bridges are highlighted in yellow, while residues involved in magnesium coordination are shown in red.

**Figure 11 toxins-17-00262-f011:**
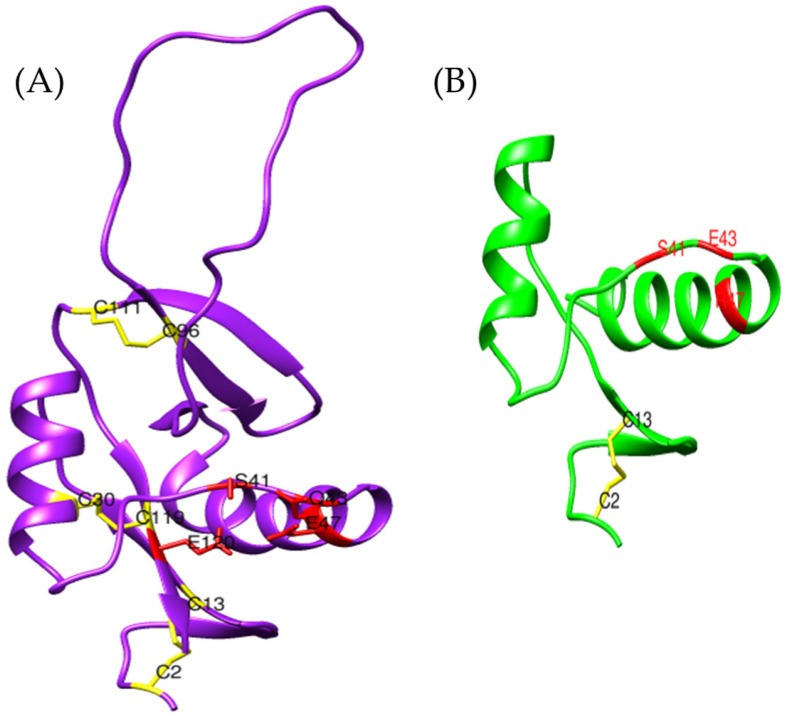
Comparison between the three-dimensional model obtained for (**A**) *Bothrops asper* (purple) using AlphaFold2 and that of (**B**) aspercetin (green) [UniProt P0DJC9]. The magnesium-binding sites are shown in red and disulfide bonds in yellow.

**Figure 12 toxins-17-00262-f012:**
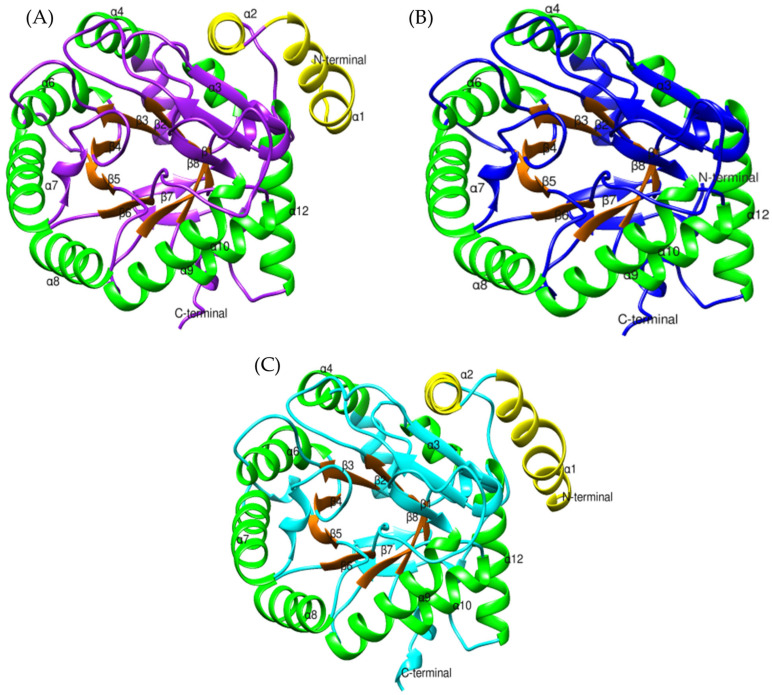
Structural comparison of dihydroorotate dehydrogenase models: (**A**) *Bothrops asper* (purple), (**B**) *Bothrops jararaca* (blue), and (**C**) *Homo sapiens* (cyan) [PDB: 3U2O]. The large C-terminal domain is shown with α-helices in green and β-strands in orange, while the small N-terminal domain is depicted in yellow.

**Figure 13 toxins-17-00262-f013:**
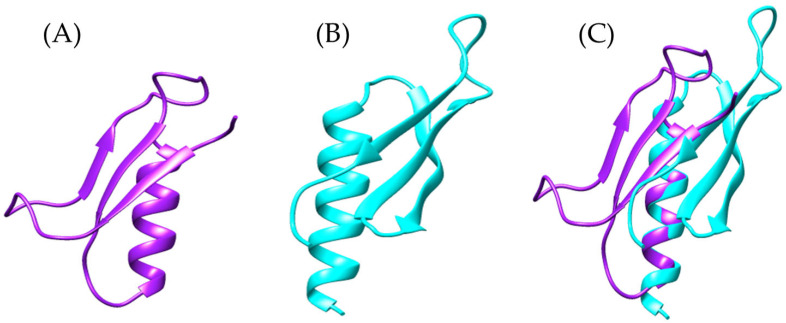
Structural comparison of the modeled proteins: (**A**) In purple, the predicted structure of the sequence identified in the *Bothrops asper* venom gland transcriptome (Ba_1), (**B**) in cyan the structure of basparin [UniProt: P84035], and (**C**) structural overlay of both models highlighting conformational differences.

**Figure 14 toxins-17-00262-f014:**
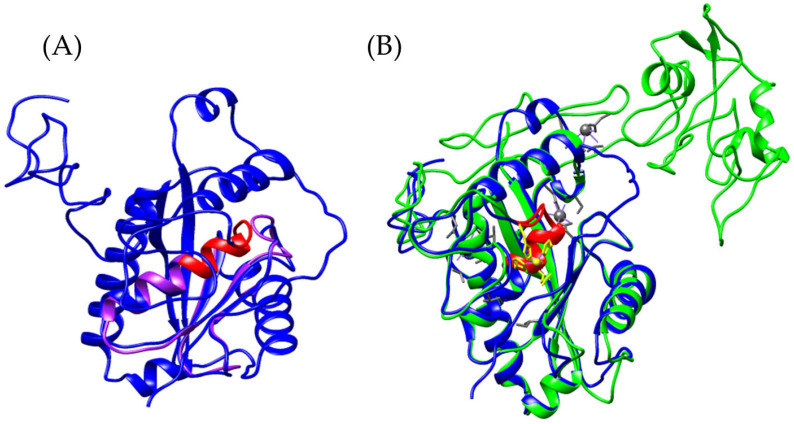
Structural identification of the zinc-binding motif (red). (**A**) Superposition of the Ba_1 model (purple) and the newly identified *Bothrops asper* variant “B_asper_bothropasin_AF” (blue). (**B**) Structural comparison between bothropasin from *Bothrops jararaca* (green) [UniProt O93523] and the B_asper_bothropasin_AF model (blue). Zinc-coordinating residues are highlighted in yellow, and calcium-binding residues in gray.

**Figure 15 toxins-17-00262-f015:**
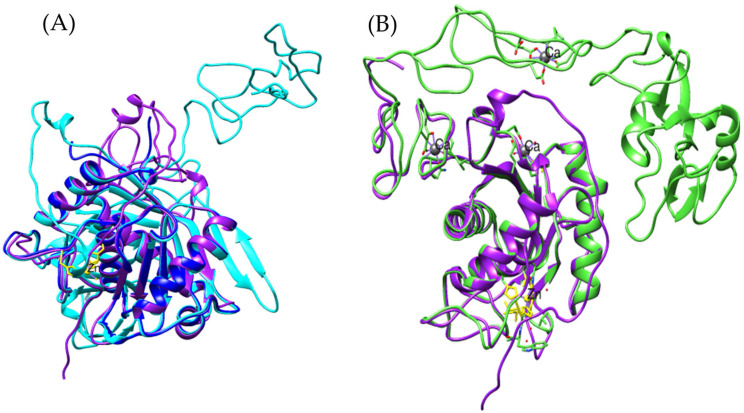
(**A**) Structural comparison of metalloproteinases from *Bothrops asper*. The B_asper_bothropasin_AF model is shown in purple, bothrasperin [UniProt: Q072L5] in cyan, and BaP1 [UniProt: P83512] in blue. Zinc- and calcium-binding residues are highlighted in yellow and gray, respectively. (**B**) Structural overlay of the B_asper_bothropasin_AF model (purple) with the VAP2B model from *Crotalus atrox* [UniProt: Q90282] (green).

**Table 1 toxins-17-00262-t001:** Structural validation of the *Bothrops asper* and *Bothrops jararaca* models.

Proteins		ERRAT	RAMACHANDRAN	Verify3D
Allergen 5	*B. asper*	95.74%	92.3%	92%
	*B. jararaca*	93.52%	90.3%	98.64%
Von Willebrand Factor	*B. asper*	84%	81.3%	48.45%
	*B. jararaca*	84.53%	88.5%	76.14%
Arylsulfatase	*B. asper*	86.62%	86.7%	73.68%
	*B. jararaca*	80.89%	88.6%	72%
Botrocetin	*B. asper*	84.34%	87%	56.91%
	*B. jararaca*	82.14%	82.8%	61.42%
Dihydroorotate dehydrogenase	*B. asper*	96.51%	92.7%	81.41%
	*B. jararaca*	99%	92.1%	93.13%
Basparin	*B. asper*	100%	82%	1.79%
Bothropasin	*B. asper*	86%	85%	59%

**Table 2 toxins-17-00262-t002:** Proteins used as reference.

Protein	Species	ID
Allergen 5	*Vespula germanica*	P35784
Apolipophorin	*Odontomachus monticola*	A0A348G622
von Willebrand factor	*Homo sapiens*	P04275
Arylsulfatase I	*Ophiophagus hannah*	V8PBU1
Botrocetin	*Bothrops jararaca*	P22030
Dihydroorotate dehydrogenase	*Homo sapiens*	Q02127
Basparin	*Bothrops asper*	P84035
Bothropasin	*Bothrops jararaca*	O93523

**Table 3 toxins-17-00262-t003:** Proteins used in comparative structural analyses.

Protein	Species	ID
Allergen 5	*Vespula germanica*	UniProt P35784
Triflin	*Trimeresurus flavoviridis*	PDB 1WVR
vWFD	*Homo sapiens*	PDB 6N29
Arylsulfatase	*Ophiophagus hannah*	UniProt V8PBU1
Arylsulfatase B	*Homo sapiens*	PDB 1FSU
Botrocetin	*Bothrops jararaca*	PDB 1FVU
Aspercetin	*Bothrops asper*	UniProt P0DJC9
Dihydroorotate dehydrogenase	*Homo sapiens*	PDB 3U2O
Basparin A	*Bothrops asper*	UniProt P84035
Bothropasin	*Bothrops jararaca*	PDB 3DSL
Bothrasperin	*Bothrops asper*	UniProt Q072L5
Snake venom metalloproteinase BaP1	*Bothrops asper*	PDB 1ND1
Zinc metalloproteinase-disintegrin-like VAP2B	*Crotalus atrox*	PDB 2DW0

Note: While some proteins included in the comparative analysis have experimentally resolved structures deposited in the Protein Data Bank (PDB), others are only available as predicted models generated by AlphaFold and retrieved from UniProt. Despite not being experimentally resolved, these AlphaFold-predicted structures exhibit a high degree of structural reliability and were therefore considered appropriate for the comparative analyses performed in this study.

## Data Availability

The data presented in the study are openly available in Zenodo repository at: https://doi.org/10.5281/zenodo.15486763 (accessed on 16 April 2025).

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
