# Peer review of "Computational Modeling of Low-Abundance Proteins in Venom Gland Transcriptomes: Bothrops asper and Bothrops jararaca"

_toxins, 2025, doi:10.3390/toxins17060262_

Round 1

Reviewer 1 Report

Comments and Suggestions for Authors

This manuscript explores and describes low abundance venom proteins derived from venom gland transcriptomic data publicly available for Bothrops asper and Bothrops jararaca (NCBI; SRR12800503 and SRR12915695).  The manuscript is well written and organized, and the figures are appropriate.  Previously, 14 'high' abundance proteins were described from these same venom transcriptomes (described in ref 19).  The current paper further explores seven (7) 'low' abundance venom proteins using similar bioinformatic tools and analysis for the following; 1. apolipophorin, 2. allergen 5, 3. arylsulfatase, 4. botrocetin, 5. dihydropyrimidine dehydrogenase, 6. aspercetin, and 7. basparin A.  The author(s) additionally provide a comprehensive literature review in the context of the bioinformatics and computational findings.   

BLASTX searches were first used to identify associated protein families from the transcriptomic RNAseq data through conserved regions and domains across species in public databases using bioinformatic tools.  The predicted protein tertiary structure for each was then examined using AlphaFold2 modeling with comparisons to reference proteins from other species.

The authors sequentially and systematically discuss each of the seven proteins and evaluate the modeled protein structures for these low abundance venom peptides.  The AlphaFold2-predicated protein structures for all seven proteins were assessed using 3 different computational structure validation models (i.e., PROCHECK, ERRAT, and Verify3D).  All AlphaFold2-predicated structures were deemed to be structurally reliable from this analysis, with some notable low scores attributed to intrinsically disordered regions withing the proteins.

Given that these low-abundance proteins cannot be effectively isolated from snake venom and characterized through conventional experimental approaches, the bioinformatics and computational modeling approach reported in this manuscript provides a rational basis for future studies that would necessarily involve generating recombinant proteins in sufficient quantities that could then be tested experimentally, including the validation of their structures (e.g., NMR) and characterizing their toxicity profiles in vitro and in vivo.

Major points. 

  1. The level of abundance for the 7 proteins compared to the previous 14 proteins from these two transcriptomes (Ref 19) are not provided, despite being described as “low abundance”. It is recommended that a table be provided, listing all 21 proteins identified from both venom gland transcriptomic data with their ranked abundance and how the relative abundance was determined or estimated.
  2. Some of the reference proteins comparisons do not appear to be with an experimentally determined PDB protein crystal or NMR structure (Table 2), but instead with another computationally based model. This should be more clearly stated in the text, noting model-to-model comparisons.

Author Response

Comments 1: The level of abundance for the 7 proteins compared to the previous 14 proteins from these two transcriptomes (Ref 19) are not provided, despite being described as “low abundance”. It is recommended that a table be provided, listing all 21 proteins identified from both venom gland transcriptomic data with their ranked abundance and how the relative abundance was determined or estimated.

Response 1: We appreciate your valuable observation regarding the inclusion and relative abundance of proteins identified in the analyzed transcriptomes. In this section, we considered both the most abundant toxin families widely reported in the literature—such as PLA2, SVMP, and SVSP—as well as proteins identified in our study that were classified as low-abundance based on their relatively low expression levels or limited representation in public databases. 

In response to your comment: We would like to clarify that this is not fully feasible due to the nature of the available data and the selection criteria applied in our study. The referenced article identifies the most abundant transcripts as corresponding to metalloproteinases (SVMP), bradykinin-potentiating peptides (BPP), venom serine proteinases (SVSP), C-type lectins (CTL), vascular endothelial growth factors (svVEGF), phospholipases A2 (PLA2), PLA2 inhibitors, cysteine-rich secretory proteins (CRISPs), and L-amino acid oxidases (LAAO). These toxin families are also represented in our abundance table.

Regarding the seven proteins specifically discussed in our manuscript, several are already included in the table: the von Willebrand factor, arylsulfatase, and botrocetin were identified with sufficient bioinformatic support. In the case of Allergen 5, the associated contig showed higher similarity to CRISP proteins, and was thus classified within that family in the abundance table. The dihydropyrimidine dehydrogenase was mentioned as a comparative reference, but the contig identified showed greater similarity to dihydroorotate dehydrogenase, which is included in the analysis. Basparin A is a protein exclusive to Bothrops asper, with only a single entry in UniProt, and due to the lack of broader representation, it was not included. Lastly, aspercetin was used solely as a structural comparison model for Bothrops asper, based on the botrocetin sequence from Bothrops jararaca. Since only a single contig was associated with this protein, and without additional support, it was not deemed appropriate for inclusion in the abundance table.

For these reasons, the final abundance table does not list exactly 21 proteins, but rather includes only those that met stringent identification criteria, totaling 15. This decision underscores the methodological rigor of our bioinformatic and structural approach. The complete abundance data are available in Supplementary Material 3 and include: Botrocetin, C-type Lectin, Cysteine-rich Protein, Dihydroorotate Dehydrogenase, L-amino Acid Oxidase, Metalloproteinase, Serine Protease, von Willebrand Factor, von Willebrand Factor Type D, Arylsulfatase, Bradykinin-Potentiating Peptide, Lysophospholipase D, Phospholipase A2, Phospholipase A2 Inhibitor, and Snake Venom Vascular Endothelial Growth Factor.

Comments 2: Some of the reference proteins comparisons do not appear to be with an experimentally determined PDB protein crystal or NMR structure (Table 2), but instead with another computationally based model. This should be more clearly stated in the text, noting model-to-model comparisons.

Response 2: We appreciate your observation regarding the content of Table 2. The primary objective of this table is to list the reference proteins used for TBLASTN searches within the Bothrops asper and Bothrops jararaca transcriptomes. Therefore, it does not refer to the proteins employed in the structural comparison analyses discussed in the “Discussion” section.

To avoid potential confusion, we have revised and clarified the “4.1 Data Acquisition” section and have modified Table 2 to display only the accession identifiers (IDs) of the protein sequences specifically used in the transcriptomic searches.

Additionally, we have included a new table, entitled “Table 3. Proteins used in the structural comparison analyses,” within section “4.3 Modeling.” This new table lists all proteins used in the structural comparisons, explicitly indicating whether the structure corresponds to an experimentally resolved protein available in the Protein Data Bank (PDB) or to a computational model predicted by AlphaFold and retrieved from UniProt.

It is important to note that while some proteins have experimentally resolved structures, others are only available as predicted models. Nonetheless, AlphaFold-predicted models are widely recognized for their high structural reliability and were thus deemed suitable for the comparative analyses conducted in this study.

Reviewer 2 Report

Comments and Suggestions for Authors

The authors show an interestion computational study of snake venom proteins. The manuscript is written very well.

My main criticism concerns title and abstract. Both are somewhat misleading as the reader expects experimental data and experimental validation and both are not provided. I suggest a title such as “Computational exploration of the tertiary structure of low-abundance proteins …. “, because this is what has been done using published genomic sequence information.

Also, the proteins are not novel at all, because their sequences can be inferred from published genomic databases.

Likewise, the abstract should be written in such a way that the computational aspect of the work becomes clear. What has not been done, is “This study identifies and characterizes novel low-abundance proteins”. The study has rather taken published sequence information, compared it to other published protein information and simulated tertiary protein structures of their proteins of choice.

The authors also talk about validation of the structures the calculated. One would expect experimental validation, but again, this was not the case. The process of validation was based entirely on different software tools.

In summary, the paper is interesting, but the authors should state clearly from the start what they have done.

Author Response

Comments 1: My main criticism concerns title and abstract. Both are somewhat misleading as the reader expects experimental data and experimental validation and both are not provided. I suggest a title such as “Computational exploration of the tertiary structure of low-abundance proteins …. “, because this is what has been done using published genomic sequence information.

Response 1: We appreciate your comment. Both the title and the abstract have been revised to avoid any ambiguity regarding the scope of the study. We have clarified that the primary objective is to analyze low-abundance proteins using computational tools, as their low expression levels currently prevent experimental validation through conventional laboratory methods. However, bioinformatic approaches allow us to predict their potential three-dimensional structures and infer possible biological functions based on the conservation of structural and functional motifs.

The revised title of the manuscript is: Computational Modeling of Low-Abundance Proteins in the Venom Gland Transcriptomes: Bothrops asper and Bothrops jararaca.

Comments 2: Also, the proteins are not novel at all, because their sequences can be inferred from published genomic databases.

Response 2: We appreciate your comment. Although the proteins analyzed in this study have been reported in other species and their sequences can be inferred from public databases, to the best of our knowledge, the specific proteins described here have not been previously reported in Bothrops asper or Bothrops jararaca. Therefore, we consider them to be novel in the context of these species, or potentially represent new isoforms.

For instance, there are no prior reports of arylsulfatase, dihydroorotate dehydrogenase, or von Willebrand factor in B. asper or B. jararaca. Among other snake lineages, only arylsulfatase has been the subject of a detailed functional analysis, specifically in Naja nigricollis. The identification and computational modeling of these proteins’ three-dimensional structures thus provide new insights into the venom gland transcriptomes of B. asper and B. jararaca.

In addition, proteins such as basparin and aspercetin are poorly represented in Bothrops asper databases. The sequences and models obtained in our study show differences from, and in some cases greater completeness than, those currently available in UniProt. These findings expand the available data and suggest the existence of potentially novel isoforms in this species.

Comments 3: Likewise, the abstract should be written in such a way that the computational aspect of the work becomes clear. What has not been done, is “This study identifies and characterizes novel low-abundance proteins”. The study has rather taken published sequence information, compared it to other published protein information and simulated tertiary protein structures of their proteins of choice.

Response 3: We appreciate your observation and agree that the proteins analyzed in this study are not novel in a general sense, as they have been previously reported in other organisms. However, this work represents the first report of arylsulfatases and dihydroorotate dehydrogenase in Bothrops asper and Bothrops jararaca.

Furthermore, in the case of proteins such as aspercetin, basparin, and bothropasin, the sequences obtained exhibit structural variations compared to previously reported sequences, while retaining characteristic domains and functional motifs. Therefore, we propose that these may represent new isoforms in these species. Accordingly, the abstract has been revised to clearly emphasize the computational nature of the study and to avoid any implication that novel proteins were experimentally identified.

Comments 4:The authors also talk about validation of the structures the calculated. One would expect experimental validation, but again, this was not the case. The process of validation was based entirely on different software tools.

Response 4: We appreciate your comment. The objective of this manuscript is to analyze low-abundance proteins that have not been studied in Bothrops asper and Bothrops jararaca, and those proteins that have few reports in databases. Furthermore, we have seen in the literature that low-abundance proteins have not been analyzed in depth, given that several authors mention the difficulty of isolating them and obtaining crystallographies using laboratory methods. Therefore, we have emphasized that the use of current computational tools allows us to characterize proteins that cannot be obtained using conventional experimental methods or that are very difficult and expensive to perform.

Reviewer 3 Report

Comments and Suggestions for Authors

1. While the manuscript thoroughly explores several low-abundance proteins, the main hypothesis or central research question is not clearly defined in the introduction. Please clearly articulate a unifying hypothesis regarding why these specific low-abundance proteins were selected and how their identification informs venom evolution, toxicity, or therapeutic applications.
2. The work is entirely in silico, relying on transcriptome data and predictive modeling tools such as AlphaFold2. While this is acceptable for preliminary characterization, the absence of even minimal experimental validation (e.g., RT-PCR confirmation or proteomics-based detection) is a significant limitation. Authors should provide supporting expression evidence (FPKM/TPM values) for each identified low-abundance transcript. Also discuss the possibility of cross-verifying the expression of selected proteins using targeted mass spectrometry in future work.
3. Some modeled proteins (e.g., basparin) showed extremely poor Verify3D scores (as low as 1.79%), raising questions about the reliability of structural inferences (Table 1; Supplementary Material 2). Authors mustclearly justify the acceptance criteria for such models. Discuss in the Results or Discussion the implications of these poor validation scores. Possibly exclude or treat with caution the conclusions based on these weak models.
4. Confusion Between CRISP and Allergen 5 Proteins
The manuscript interchanges the terms Allergen 5 and CRISP, particularly in the Antigen 5 section (2.1), without clearly resolving the functional distinction or confirming sequence identity. Given that CAP domain proteins are diverse, please clarify whether the protein modeled is better classified as CRISP or Ag5, based on sequence homology and phylogenetic clustering. Include a phylogenetic tree (if available) to justify this classification.
5. The functional implication of identified proteins is speculative. Many functional interpretations (e.g., zinc-mediated inflammatory responses, Figure 3; prothrombin activation) are highly speculative and based on analogy rather than direct evidence. Authors should temper the language in the Discussion sections to clearly indicate which inferences are hypothetical. 
If possible, cite venom studies or proteomics datasets showing these proteins in active venom to support the discussion.
6. Domain-based structural interpretations need strengthening. The comparative structural modeling using AlphaFold2 is extensive, but domain predictions and alignments would benefit from better visual representation. Include annotated domain architecture diagrams for each protein showing the predicted domains across species (Bothrops, reference species, human, etc.). 
Supplementary Figure legends (e.g., FS3–FS17) lack adequate explanations. Please provide clear, standalone captions.
7. Several inconsistencies were observed in protein nomenclature and terminology. The manuscript refers to the same protein under different names (e.g., “basparin” and “Ba_1”) without a clear naming convention. Ensure consistent protein naming throughout text, figures, tables, and supplementary files. Include a table summarizing all identified proteins, their source contigs, predicted domains, and classification (CRISP, SVMP-PIIIb, etc.).
8. Comparative evolutionary insight is lacking. The potential evolutionary significance of low-abundance proteins is hinted at but not fully developed. Discuss how these proteins might contribute to species-specific venom phenotypes or ecological adaptations. Reference additional viperid or elapid studies that include low-abundance protein characterization.

Comments on the Quality of English Language

The manuscript contains repetitive phrasing and long paragraphs that affect readability. For instance, multiple mentions of “suggesting potential structural conservation” or “conserved motifs” can be tightened. Ensure consistent referencing format.

Author Response

Comments 1: While the manuscript thoroughly explores several low-abundance proteins, the main hypothesis or central research question is not clearly defined in the introduction. Please clearly articulate a unifying hypothesis regarding why these specific low-abundance proteins were selected and how their identification informs venom evolution, toxicity, or therapeutic applications.

Response 1: We appreciate your observation regarding the need to more clearly define the central hypothesis and research question. In the revised version of the manuscript, we have restructured the introduction to explicitly state the study's objective and provide a rationale for selecting low-abundance proteins.

This study focuses on the identification and structural modeling of poorly characterized proteins present in the venom gland transcriptomes of Bothrops asper and Bothrops jararaca, aiming to infer their potential roles in venom toxicity. The central hypothesis posits that, despite their low expression levels, these proteins may perform biologically relevant functions due to the conservation of structural domains associated with toxic activity or immune evasion mechanisms.

Furthermore, we clarify that this is a computational study, given the experimental challenges in isolating these proteins. The bioinformatic approach enables a preliminary characterization that can serve as a foundation for future functional validation through synthetic or heterologous expression systems. We have also removed the term “venom evolution” to avoid misinterpretation, as no phylogenetic analysis is included. Instead, we emphasize the relevance of conserved structural features as a more appropriate and consistent framework for this study.

Comments 2: The work is entirely in silico, relying on transcriptome data and predictive modeling tools such as AlphaFold2. While this is acceptable for preliminary characterization, the absence of even minimal experimental validation (e.g., RT-PCR confirmation or proteomics-based detection) is a significant limitation. Authors should provide supporting expression evidence (FPKM/TPM values) for each identified low-abundance transcript. Also discuss the possibility of cross-verifying the expression of selected proteins using targeted mass spectrometry in future work.

Response 2: We appreciate your comment regarding the importance of supporting the expression of low-abundance proteins with quantitative evidence. Although our study is based exclusively on in silico analysis, we have incorporated a robust estimate of relative expression using TPM values ​​generated with Salmon, a high-precision quantification tool for transcriptome data.

To document this information clearly and transparently, we have added two levels of complementary evidence:

a. Summary table by protein (Table 1):
Includes total TPM values, the cumulative number of mapped reads, and the relative percentage of expression of each low-abundance protein relative to the entire transcriptome. Proteins such as von Willebrand factor type D, arylsulfatase, and dihydroorotate dehydrogenase are highlighted, showing low but detectable expression levels.

b. Detailed tables by protein (Tables S1–S5):
Contain all contigs associated with each protein, with their respective individual TPM values, number of reads, and effective length. This disaggregation allows for the assessment of expression consistency across multiple transcripts.

We recognize that the low abundance of these transcripts requires cautious interpretation. However, their reproducible detection in different contigs, along with the conservation of functional domains, provides a reasonable basis for their inclusion in functional analysis. Finally, we have incorporated into the discussion the possibility of experimentally validating these proteins in future studies using techniques such as RT-PCR or targeted proteomics, which would further strengthen the findings obtained in this exploratory phase.

Comments 3: Some modeled proteins (e.g., basparin) showed extremely poor Verify3D scores (as low as 1.79%), raising questions about the reliability of structural inferences (Table 1; Supplementary Material 2). Authors mustclearly justify the acceptance criteria for such models. Discuss in the Results or Discussion the implications of these poor validation scores. Possibly exclude or treat with caution the conclusions based on these weak models.

Response 3: Thank you for your comment. It is true that the model generated for basparin A showed a low Verify3D score (1.79%), which may raise concerns regarding its structural reliability. However, when compared with the original basparin A protein sequence used as a reference, we observed that it also displayed a moderately low Verify3D score (55.56%), with several regions falling below the validation threshold. This pattern suggests that the low scores are likely due to the presence of flexible or intrinsically disordered regions, which are notoriously challenging to model and validate using computational tools, and do not necessarily imply an incorrect or unstable model.

Despite the low Verify3D score, other validation metrics support the structural plausibility of the model: 82% of residues are located in favored regions of the Ramachandran plot, and 100% of the structure falls within the acceptable range according to the ERRAT quality factor. Therefore, we chose to retain the model for subsequent analyses, while interpreting its results with appropriate caution. This limitation has been acknowledged in the Discussion section under “2.7. Model Validation,” and an explanatory note has been added to Supplementary Material 2.

Comments 4: Confusion Between CRISP and Allergen 5 Proteins
The manuscript interchanges the terms Allergen 5 and CRISP, particularly in the Antigen 5 section (2.1), without clearly resolving the functional distinction or confirming sequence identity. Given that CAP domain proteins are diverse, please clarify whether the protein modeled is better classified as CRISP or Ag5, based on sequence homology and phylogenetic clustering. Include a phylogenetic tree (if available) to justify this classification

Response 4: We appreciate your comment regarding the confusion between CRISP proteins and Allergen 5. After a detailed review of the functional annotation and structural modeling, we confirmed that the analyzed sequence shows greater homology with proteins from the CRISP (Cysteine-Rich Secretory Proteins) group than with allergens type 5. Consequently, the text has been corrected to refer exclusively to this protein as CRISP.

Although this study does not include a formal phylogenetic analysis of all CAP domain members, the classification is supported by the results obtained from BLAST searches and alignment analyses performed with Jalview, which show greater similarity with CRISP proteins present in snake venoms. Furthermore, the title of section 2.1 has been modified, and any reference to “Allergen 5” in relation to this sequence has been removed.

Regarding the inclusion of a phylogenetic tree, although it was not incorporated in this version of the manuscript due to the focus on structural and functional analysis, we recognize its relevance and consider it a priority line for future studies that delve into the evolutionary relationships within the CAP domain.

Comments 5:  The functional implication of identified proteins is speculative. Many functional interpretations (e.g., zinc-mediated inflammatory responses, Figure 3; prothrombin activation) are highly speculative and based on analogy rather than direct evidence. Authors should temper the language in the Discussion sections to clearly indicate which inferences are hypothetical. 
If possible, cite venom studies or proteomics datasets showing these proteins in active venom to support the discussion.

Response 5: We appreciate your comments regarding the functional interpretation of the identified proteins. We fully agree that it is essential to clearly distinguish between observations supported by direct evidence and those based on functional inference. In our manuscript, the proposed hypotheses are grounded in previous studies that have reported these proteins in active venoms of other viper species, particularly through proteomic analyses. These studies support the functional presence of homologous proteins, as well as the biological relevance of certain domains, conserved motifs, and characteristic structures.

Nevertheless, we have carefully considered your observation and have revised several sections of the Discussion to explicitly indicate which statements are based on hypothetical inferences. Specifically, we have moderated the language in the following sections:

Lines 190–194: "Therefore, we suggest the possibility that the CRISPs identified in Bothrops asper and Bothrops jararaca could play a similar role, favoring the activation of inflammatory responses through zinc-mediated mechanisms. This effect could have implications in local inflammation after the bite, contributing to the alteration of the endothelium, which could enhance the tissue damage observed in bothropypic envenomation [26, 34, 38]."

Lines 254-261: "Despite the structural differences, the identification of a sequence with similarity to the vWFD domain in B. asper suggests a potential role in the interactions with coagula-tion factors such as FVIII and vWF. This is consistent with the known protein diversity of Bothrops venom, where proteins with similar functions have been previously de-scribed. Toxins such as bitiscetin and bothrocetin have been reported in certain species of this genus, capable of modulating prey coagulation by interacting with these factors and enhancing the hemorrhagic effects of the venom [46], [47], [48], [49], [50], [51], [52]."

Lines 339-346: "The conservation of Cys91 in the arylsulfatase of Bothrops asper may indicate a role in the degradation of sulfated biomolecules such as a glycosaminoglycans, ten-dons, or ligaments a process demonstrated in the venoms of other species, as discussed. Since Cys91 has been implicated in the stabilization of sulfate esters, its presence un-derscores its potential functional relevance in enzymatic activity and its possible in-volvement in the toxic or digestive processes of B. asper. These findings suggest that the identified sequences may contribute to the venom’s toxicity mechanisms by promoting interactions with specific biological targets and enhancing the effects of other toxins."

These modifications aim to clarify that, while our functional proposals are supported by previous scientific literature, the specific roles in Bothrops asper and Bothrops jararaca remain hypothetical.

Comments 6: Domain-based structural interpretations need strengthening. The comparative structural modeling using AlphaFold2 is extensive, but domain predictions and alignments would benefit from better visual representation. Include annotated domain architecture diagrams for each protein showing the predicted domains across species (Bothrops, reference species, human, etc.). 
Supplementary Figure legends (e.g., FS3–FS17) lack adequate explanations. Please provide clear, standalone captions.

Response 6: We appreciate your comment. In response, we have included in Supplementary Material 1 the domain annotation diagrams generated with PFAM for each of the analyzed proteins. They are available in PNG format in the folder for each protein. These diagrams clearly visualize the domain architecture and functional regions identified in our sequences, facilitating structural comparisons between species.

Furthermore, we would like to point out that the multiple alignments presented for each protein already included the positions of the domains and conserved sites recovered from PFAM. However, we agree that the individual PFAM diagrams offer a more direct and understandable visualization of the structural organization of each protein, and we believe they effectively complement the analyses already included.

Finally, we have revised and expanded the legends in Supplementary Figures FS3–FS17 to ensure that they are clear, self-contained, and provide the necessary context for correct interpretation.

Comments 7: Several inconsistencies were observed in protein nomenclature and terminology. The manuscript refers to the same protein under different names (e.g., “basparin” and “Ba_1”) without a clear naming convention. Ensure consistent protein naming throughout text, figures, tables, and supplementary files. Include a table summarizing all identified proteins, their source contigs, predicted domains, and classification (CRISP, SVMP-PIIIb, etc.).

Response 7: We appreciate the suggestion to include a summary table listing the identified proteins, their corresponding contigs, predicted domains, and classification. However, the functional information retrieved for the proteins is highly variable: while some sequences are associated only with representative families, others contain well-defined domains, conserved motifs, or active sites. Due to this heterogeneity, we believe that a general table would not adequately reflect the level of detail available for each protein.

For this reason, we chose to present this information in a more comprehensive and visual manner through the PFAM-generated graphics included in Supplementary Material 1. These figures clearly depict the domains, families, and conserved sites for each analyzed protein. Additionally, we have complemented this visualization with the PFAM tabular output (Supplementary File), which includes sequence information and the precise positions of the domains, motifs, and functional regions identified in the proteins described in the manuscript.

We have also clarified the nomenclature used in the study. Specifically, "Ba_1" refers to the sequence obtained in Bothrops asper using Basparin as a reference. This is now explicitly stated in lines 503–505 of the manuscript:
“For this reason, Basparin was used as a reference to search for it in the transcriptome of the venom gland of Bothrops asper, leading to the identification of a matching sequence, here referred to as Ba_1.”

A similar clarification was made for the sequence identified using bothropasin (lines 530–534):
“To test this hypothesis, bothropasin from Bothrops jararaca was used as a reference in the analysis of the Bothrops asper transcriptome. This approach yielded a sequence similar to Ba_1 (Figure 14A), but considerably larger, which we designated as ‘B_asper_bothropasin_AF’ to avoid confusion.”

Comments 8: Comparative evolutionary insight is lacking. The potential evolutionary significance of low-abundance proteins is hinted at but not fully developed. Discuss how these proteins might contribute to species-specific venom phenotypes or ecological adaptations. Reference additional viperid or elapid studies that include low-abundance protein characterization.

Response 8: We appreciate your comments regarding the comparative evolutionary perspective. In the manuscript, we addressed how low-abundance proteins may contribute to venom diversification in Bothrops asper and Bothrops jararaca. Although dihydroorotate dehydrogenase (DHODH) and the von Willebrand factor type D domain-like protein (vWFD-like) have not been previously reported in viperid venoms or in other venomous organisms, their presence in the transcriptome suggests potential functional recruitment events with possible evolutionary implications. To explore this hypothesis, we compared their conserved domains and structural models with homologous proteins in mammals, which allowed us to infer potential roles in metabolic and coagulation-related processes. Despite the lack of prior studies in snakes, comparisons with other organisms provide a valuable perspective.

Furthermore, we have highlighted how proteins such as arylsulfatase, CRISP, basparin, and botrocetin previously reported in other viperid and elapid species may be involved in the functional diversity of the venom. In this context, we integrated data from other species to compare the roles of minor venom components in envenomation.

This approach provides an initial framework for future comparative and evolutionary studies aimed at further exploring the specificity and potential roles of these proteins within the context of snake venom systems.

Comments 9: The manuscript contains repetitive phrasing and long paragraphs that affect readability. For instance, multiple mentions of “suggesting potential structural conservation” or “conserved motifs” can be tightened. Ensure consistent referencing format.

Response 9: We appreciate your comment regarding repetitive phrasing and paragraph structure. We acknowledge that expressions such as “suggesting potential structural conservation” or “conserved motifs” appear multiple times throughout the manuscript. However, this repetition is intentional, as each protein is analyzed individually with respect to similar characteristics such as domains, conserved motifs, and functional structures. Therefore, while certain terminology is repeated, we believe it is necessary to preserve scientific accuracy and consistency in the discussion of each case

Round 2

Reviewer 2 Report

Comments and Suggestions for Authors

ok

Reviewer 3 Report

Comments and Suggestions for Authors

Authors addressed the queries raised.